# Considerations for the Prosthetic Dental Treatment of Geriatric Patients in Germany

**DOI:** 10.3390/jcm10020304

**Published:** 2021-01-15

**Authors:** Ina Nitschke, Anja Wendland, Sophia Weber, Julia Jockusch, Bernd Lethaus, Sebastian Hahnel

**Affiliations:** 1Clinic for Prosthetic Dentistry and Dental Materials Science, Leipzig University Medical Center, 04103 Leipzig, Germany; ina.nitschke@medizin.uni-leipzig.de (I.N.); sophia.weber@medizin.uni-leipzig.de (S.W.); 2Clinic of General, Special Care and Geriatric Dentistry, University of Zurich, 8006 Zurich, Switzerland; julia.jockusch@zzm.uzh.ch; 3Department of Internal Medicine and Geriatrics, Barmherzige Brüder Hospital, 93049 Regensburg, Germany; anja.wendland@barmherzige-regensburg.de; 4Clinic for Oral, Maxillofacial, and Plastic Surgery, Leipzig University Medical Center, 04103 Leipzig, Germany; bernd.lethaus@medizin.uni-leipzig.de

**Keywords:** gerodontology, geriatric assessment, prosthetic dentistry, implant, denture, overdenture

## Abstract

Demographic changes in the industrialized countries require that dentists adapt to the growing and heterogeneous group of elderly patients and develop concepts for the dental care of fit, frail, and dependent old and very old people. In general, dental care for old and very old people should be based on their individual everyday life. As a result of demographic changes, improved oral hygiene at home, and the establishment of professional teeth and denture cleaning, tooth loss occurs increasingly in higher ages, which implies that first extensive prosthetic rehabilitation with fixed or/and removable dental prostheses is shifting to a higher average age than ever before. This phenomenon requires that the individual diseases, potential multimorbidity and polypharmacy, and associated limitations are taken into consideration. Against this background, the current survey aims to summarize epidemiological trends associated with tooth loss, using Germany as a highly representative country for demographic changes as an example. Furthermore, the current narrative summary outlines general principles that should be followed in dental care, treatment of geriatric patients, and outlines current therapeutic options in prosthetic dentistry.

## 1. Epidemiology and General Aspects

To date, Germany features a society with one of the world’s oldest populations. Demographic analyses issued by the German Federal Institute for Population Research indicate that the number of inhabitants aged 65 years and older will increase to 28% in 2040 and 30% in 2060 [1], which is a phenomenon that is similar in numerous other industrialized countries. In addition to the increasing number of old and very old people, the number of people in Germany receiving nursing care is also increasing, reaching more than 3.4 million in 2017 [2].

With regard to the dental status, epidemiological studies highlight that the dental status in old and very old people has changed in recent years. In German seniors aged 65–74 years, the average number of missing teeth (excluding the third molars) declined from 21.9 in 1997 over 17.8 in 2005 to 14.6 in 2014 [3,4,5]. Still, more than 35% of German seniors aged 65–74 years are supplied with removable dental prostheses (RDPs) [5]. For the Fifth German Oral Health study in 2014, older seniors aged between 75–100 years were included in the evaluations for the first time. This cohort featured an average number of remaining 10.2 teeth (excluding the third molars), yet 44.3% suffered from severe periodontitis according to Centers of Disease Control/American Association of Periodontology (CDC/AAP) guidelines [6]. Moreover, edentulism was identified in 47.1% of the upper jaws and in 34.4% of the lower jaws in the cohort of older seniors [7], while the prevalence of edentulism was 52% among people requiring care [8]. With regard to this aspect, the proportion of edentulous younger seniors aged between 65–74 years has reduced from 24.8% in 1997 over 22.6% in 2005 to 12.4% in 2014 [3,4,5]. Recent extrapolations suggest that the prevalence of edentulism is further decreasing to 4.2% in German seniors aged between 60–80 years by 2030 [9]. These considerations highlight that seniors in Germany have more natural teeth than in the past, yet are frequently affected by severe periodontitis and often supplied with RDPs.

It is clear that these demographic developments have a relevant impact on dentistry and dental treatment options. As commonly observed in old and very old people, the number of general diseases increases with age, as does the number of drugs regularly consumed. Data from Germany underline that 75.8% of female seniors aged between 65–74 years suffer from at least two chronic diseases, rising by another 6% in those older than 74 years [10]. The prevalence of polypharmacy (i.e., patients with a minimum of five concurrently used medications) ranges around 50% in the cohort of seniors aged between 70–79 years [11]. Analyses of data gathered by a major German health insurance provider have, however, highlighted that the percentage of insurance holders requiring dental care decreased from approximately 80% in those aged 75–79 years to less than 60% in those aged 90 years or older [12], although more than 98% of the seniors aged 80 years and older see a physician on a regular basis [13]. In Germany, about 90% of the population is covered by the statutory health insurance, which also (partially) covers basic dental treatment, such as extractions, root canal treatment, as well as fixed and removable denture prostheses. Dental care and treatment are usually performed in an outpatient setting in dental offices. Thus, it is likely that home visiting patients will gain importance in the future in order to secure access to dental care treatment for the aging population. With regard to this aspect, since 2014, dentists in Germany may enter contracts with health care facilities to secure regular and timely dental treatment of residents living in long-term care/health care facilities, which is rewarded by the statutory health insurance funds with higher remuneration. Since 2019, these cooperation agreements have become mandatory [14], which does, however, not impair the individual right of a patient to select a dentist of his or her choice. Long-term care facilities must give residents access to regular checkups, but it is still up to the residents or their relatives/caregivers or legal guardians to decide whether to take part in these programs or not [14].

Thus, in patients with an increased grade of frailty, geriatric dentistry can be considered to be dental care rather than dental treatment. Geriatric dentistry for patients in need for care is a branch of gerodontology. Due to the ongoing shift of tooth loss into higher ages, prosthetic treatment will be more frequently required in seniors rather than adults, and with the decreasing prevalence of edentulism, more complex prosthetic restorations such as tooth-supported fixed and RDPs will be demanded in old and very old cohorts.

Against this background, the aim of the current narrative review is to outline current pathways in geriatric dentistry from a prosthetic perspective.

## 2. Oral Geriatric Assessment

Any dental care and treatment should commence with profound diagnostics. Regarding treatment of elderly patients, conventional diagnostic means should, however, be completed by specific tools derived from geriatric assessments. In a geriatric setting, typical tools of a multidimensional and multidisciplinary geriatric assessment include, for example, the timed “Up and Go” test [15], the “Clock Completion” test [16], measurement of grip strength [17], or the “Mini Mental State” [18]. These approaches are employed to estimate the cognitive, functional, physical, and mental abilities as well as socioenvironmental settings. While the majority of instruments commonly employed in geriatrics are unsuitable for dental purposes, it is helpful to group the older patient requiring oral care in accordance with the oral functional capacity (OFC, Table 1).

OFC can be employed to assess patients from the multifactorial perspective of a dentist specialized in geriatric dentistry, taking into account a variety of aspects that may have an impact on dental care and treatment. OFC includes three parameters, i.e., therapeutic capability, oral hygiene ability, and self-responsibility. Table 2 displays the sub-points assessed for each individual parameter.
Therapeutic capability includes an estimation to what extent dental diagnosis and treatment can be performed without distinct restrictions. For instance, in a setting where dental care takes place in a nursing home or at the patients’ home, X-ray investigations are usually not possible, which limits diagnostic options and relevantly affects dental treatment. The therapeutic capability also describes how far geriatric patients can be forwarded to conventional dental treatment, including, for example, whether it is possible that the patient can reliably open the mouth for extended periods or can be positioned on a dental chair. Moreover, difficulties in adaptation toward new dental prostheses may also be assessed within this parameter.Oral hygiene ability includes aspects of oral hygiene, e.g., whether oral hygiene measures can be carried out by the patients themselves or whether a third-party cleaner is required to provide reliable support or to perform the procedure as whole. It should also be checked to ensure that patients have reliable access to oral hygiene products.Self-responsibility includes aspects such as whether patients are capable on acting on their own volition and organize themselves, or whether there is a career or a legal guardian. The issue of self-responsibility is particularly relevant for implementing a participatory decision-making process [21].

## 3. General Aspects of Dental Care and Treatment in Elderly Patients

The number of general diseases increases with age as does the number of drugs regularly consumed, which coincides with the onset of oral diseases and symptoms that are typical for older people, such as dry mouth. Moreover, difficulties in dental routine associated with the treatment of older people, including physical (such as frailty, loss of mobility, hear loss, presbyopia, or impaired manual skill and hand grip strength) or cognitive impairment (such as memory loss, dementia, or Alzheimer’s disease), underline the demand for dentists and dental personnel that are trained in dealing with old and very old patients in a dental setting. Depending on the individual definition of old and very old people, the term may as well include younger seniors aged between 65–74 years as well as very old seniors aged over 100 years, the so-called long-lived. Thus, the challenge for dentistry is to develop concepts for dental care that allow reliable dental care of people with different and changing OFCs over a period of four decades. It is clear that this circumstance goes along with very heterogeneous demands regarding dental treatment, care, as well as design and equipment of the dental office.

Therefore, the requirements for dental care and treatment of geriatric patients—taking into account the patient’s individual OFC and possible limitations in clinical/radiological diagnostic procedures—may differ from objectively required dental care and treatment (as they result from clinical and radiological diagnostics alone). With regard to this aspect, theoretical objective treatment requirements derived from clinical and radiological findings have to be transformed into a relativized and holistic treatment approach. It is clear that this approach must be based on participatory decision-making, which is particularly relevant in a geriatric setting with patients that frequently feature reduced self-responsibility (OFC). This process defines the “patients as informed consumers, helping them to make clinical decisions that optimize their personal oral health” [22]. As it has been highlighted that participatory decision-making is frequently neglected in geriatric dentistry, it should be taken into consideration that informed consent in dentistry consists of five basic elements, including:-the physical and cognitive ability to fully participate in the informed consent process,-disclosure of information associated with dental treatment (including diagnosis, risks, and benefits of treatment),-the patient’s active participation in the participative decision-making process (including verification whether the patient has heard and understood the information),-voluntariness to make his or own decision (i.e., supportive but not determinative family involvement, for instance),-final decision or choice (orally or in writing) [23].

Thus, any dental treatment plan should be based on the evaluation of OFC and participatory decision-making, which includes (final) discussion with the patients or—if necessary—their relatives or legal guardian. Close cooperation with general practitioners and geriatricians may also be helpful for improving the participatory decision-making process (for instance for evaluation of capacity) as well as dental care and treatment. Primarily, to achieve participatory involvement of patients in dental prevention and therapy, a relationship of trust in cooperation with caregivers is necessary [24]. Even in cases where patients themselves are not capable of making decisions, a therapeutic decision without information is both legally and ethically unacceptable. Here, caregivers and legal guardians, together with the dentist, must make decisions based on the presumed will of the patient [21].

## 4. Prosthetic Care and Treatment in Geriatric Patients

Epidemiological data suggest that—despite improvements in oral health in the recent decades—extensive tooth loss is still common in geriatric patients, which is why extensive RDPs still have a high prevalence in this cohort. Current concepts for the design and retention of tooth-supported RDPs that are pursued in Germany include clasp- and double crown-retained RDPs, overdentures supported by root caps, and RDPs retained by extracoronal attachments with/without crowns. It has to be taken into considerations that these concepts may further be subdivided into very distinct treatment approaches differing, for instance, on the type of retentive element employed (e.g., for double crowns: galvanic, conical, or telescopic double crown with/without clearance fit) or the individually employed restorative material (precious/non precious alloy, ceramic, polymer), which relevantly affects the individual success and failure rates.

To date, several studies have highlighted the relevance of prosthetic restorations in edentulous or partially edentulous patients for improving oral health-related quality of life [25] or decreasing the risk of the onset of cognitive disorders during the aging process [26]. In addition to that, malnutrition is a frequently observed condition in geriatric cohorts that affects up to 50% of hospitalized seniors [27]. It may be responsible for impaired functionality and quality of life and may foster morbidity as well as mortality. Regarding dentistry, it has been reported that problems associated with chewing or insufficient oral hygiene are relevant etiological agents for malnutrition [28,29], which underlines the significance of oral health and oral prevention-oriented and proactive rehabilitation in elderly patients. Moreover, catabolic metabolism in hospitalized geriatric people, often accompanied by severe general diseases, can lead to sarcopenia with atrophy of body muscle mass. In addition to that it should be borne in mind that in cases where hospitalized patients are not capable of wearing their RDPs over longer periods, alterations in anatomy may account for impaired fitting of RDPs, which might induce a circulus vitiosus fostering malnutrition. Adequate and timely dental care and treatment is mandatory in these settings.

However, as a result of the aging process and, finally, senescence, adaptation to novel prosthetic restorations may be severely impaired. To improve acceptance of prosthetic treatment or treatment outcomes in general [30], participatory decision-making regarding prosthetic treatment should be performed whenever possible. While prosthetic treatment in geriatric patients follows the common fundamental principles of prosthodontics, it is also highly individual and dependent on the distinct setting and circumstances. The patient’s individual OFC should be determined and used as a guideline for avoiding overtreatment. With regard to this aspect, any prosthetic treatment in geriatric dentistry should be based on the g-3-S principle for geriatric patients—simple, stable, and solid:Regarding insertion and removal of RDPs, these should be simple to handle for patients as well as third parties, relatives, or caregivers that might be involved in oral care.RDPs should feature a stable design to avoid fractures in case of, e.g., accidental dropping.Planning of prosthetic treatments should be solid, implying that no dental problems are to be expected in near future after completing prosthetic rehabilitation.

It is recommended to design prosthetic constructions in a manner that they can be easily modified in case of tooth loss or other biological complications. In addition to that any dental prosthesis—fixed or removable—should be easy to clean, which is particularly true in the field of gerodontology. However, RDPs feature an extensive interface that is prone to microbial adherence and plaque formation that may foster the onset of caries, periodontitis, or stomatitis. Thus, RDPs in old and very old people should be polished well and fabricated by the dental technician without extensive anatomic imitations such as alveolar yokes or plicae that promote accumulation of plaque. While the role of Candida albicans in biofilms on RDPs for the etiology of denture stomatitis has extensively been addressed [31,32,33], only little attention has so far been paid to respiratory pathogens associated with respiratory diseases such as pneumonia, which is the major cause of death in institutionalized elderly people [34]. Recent reports suggest a correlation between bacterial pneumonia and oral candidiasis [35], and RDPs may serve as a reservoir for respiratory pathogens [36]. With regard to this aspect, data gathered in community-dwelling elderly aged 85 years or older underline that the prevalence of pneumonia in persons wearing RDPs at night was associated with a 2.3 fold increased risk of the incidence of pneumonia [37]. These considerations advocate meticulous oral hygiene in elderly patients, regardless of who carries it out and who is responsible for it. As oral hygiene may be compromised in elderly patients with cognitive or motoric impairment, proper and regular instructions should be given and demonstrated not only for patients but also for caring relatives or caregivers. Scientific data proof the relevance and efficacy of continuous education in oral hygiene for caregivers [38,39]. In addition to that, insertion and removal of RDPs should be trained with both patient and caregivers. Buttons or grooves in the vestibular supporting areas of RDPs may help to improve handling. For institutionalized patients, individual labeling of RDPs can also be useful to avoid loss or confusion.

In addition to these general considerations associated with prosthetic rehabilitation in a geriatric context, some specific prosthetic treatment concepts have been developed in the past and their application in a geriatric setting is frequently discussed. Against this background, the concepts associated with shortened dental arch (SDA), implant dentistry, and duplicate dentures are introduced and discussed from a gerodontological point of view.

### 4.1. Shortened Dental Arch (SDA)

The SDA concept is a frequently discussed treatment option not only in geriatric dentistry. While adaptation to extensive RDPs can be complex and impaired particularly in older patients, SDA features a concept that avoids extensive restorations replacing molars in clinical settings with typically ten occluding antagonistic pairs (usually ranging to the second premolar). Systematic reviews and meta-analyses underline that the SDA concept is at least as successful in improving oral health-related quality of life as conventional RDPs [40], although masticatory performance is relevantly reduced and could be increased by RDPs [41]. The latter aspect should, however, not be neglected in a cohort of patients with an increased risk of sarcopenia.

For seniors aged 65 years and older, a recent randomized controlled clinical trial reported that the SDA concept was significantly more successful in comparison to rehabilitation with RDPs [42]. It is frequently discussed whether SDA concepts foster the onset and progress of temporomandibular disorders, yet, to date, no correlation has been verified [43,44]. As it has been highlighted that up to 50% of patients no longer used their RDP within five years after insertion [45], the SDA concept might be worth following, particularly in a geriatric setting and in patients without subjective demand for prosthetic treatment and without later expected functional impairment (such as elongations of antagonistic teeth). Moreover, it can be discussed that both posterior natural and denture teeth may serve as a guide rail for the bolus that are relevant for the swallowing process, which might be important in geriatric patients with dysphagia.

### 4.2. Implant Dentistry

In recent decades, implants have become an important and frequently employed treatment option in prosthodontics. While age is not regarded as a contraindication for the insertion of implants per se and scientific data indicate that the number of implants inserted in elderly patients is increasing [46]. Epidemiological surveys report that still, few missing teeth in older seniors in Germany (75–100 years old) are supplied with implants, ranging around 0.27 missing teeth replaced with implants (65–74 years: 0.22 missing teeth) [5,7].

However, it has been reported that institutionalized elderly people are rather reluctant as regards implant dentistry and 2/3 rejected the insertion of implants at all, which has predominantly been attributed to high costs associated with treatment and fears associated with implant surgery [47]. These considerations underline the relevance of OFC and participatory decision-making and indicate that surgical protocols employed in implantology in older patients should be carefully selected in dependence on the existing general diseases as well as medication and performed with minimally invasive approaches, avoiding augmentations and extensive flaps. A minimum of four implants has been advocated for rehabilitation of the edentulous maxilla, which coincides with favorable rates for implant survival [48]. For the edentulous mandible, several implantological treatment concepts have been proposed, including one, two, four, or six implants, although the options with two and four implants are the most frequently employed options. Single implants in the edentulous mandible have also been recommended as simple and cost-effective treatment option that improves denture satisfaction in elderly patients [49]. However, regarding implant survival in the edentulous mandible, data suggest that survival rates decrease with decreasing number of implants [48].

Concerning prosthetic concepts associated with the insertion implant-supported dentures, guidelines adhere to the fundamental principles applied for tooth-supported restorations, and, also in implant dentistry, prosthetic construction and denture design should adhere to the g-3-S principle, too. Nevertheless, it is well known that implants and implant-supported restorations require meticulous oral hygiene including regular proper biofilm and plaque removal to minimize the risk of periimplant infections. Thus, in a geriatric setting, it is crucial to secure oral hygiene and aftercare, which includes education of the patient, and in some cases, that of caring staff or family members. In Germany, current prosthetic concepts for extensive full-arch rehabilitation with implants include fixed restorations (fixed dental prostheses (FDPs); such as screw-retained “all-on-four” or “all-on-six” options) as well as removable restorations supported by a various number of implants. A recent systematic review indicated that FDPs feature implant survival rates in excess of 95% [50]. For screw-retained FDPs, less biological and more technical complications have been reported than for cemented FDPs, and screw-retained FDPs have been regarded as the option of choice as they can be more easily retrieved [51]. High survival rates have been reported for both implant-supported FDPs and RDPs in the upper and lower jaw, while for the lower jaw RDPs produced higher survival rates in settings with four implants than FPDs [48].

In settings with few implants (one/two), RDPs are usually designed as overdentures that are retained by bars or stud attachments such as balls or inserts (such as Locator^®^). RDPs supported by four or more implants are commonly retained by bars (prefabricated/milled) or double crowns. With regard to the different prosthetic options available, it has been reported that for RDPs splinting of implants (such as in bar-retained RDPs) does not have a significant effect on implant survival and patient satisfaction; however, unsplinted constructions (such as in double crown-retained RDPs) require more prosthetic maintenance [52]. The conventional wisdom is that fixed dental prostheses supported on implants are more difficult to clean and make high demands on the motoric skills, which is why seniors aged between 60 and 70 years preferred removable implant-supported dentures over fixed implant-supported restorations in a crossover trial [53]. Data indicate that patient satisfaction and chewing efficiency are almost similar between fixed and removable implant-supported dentures [53,54]. However, in comparison to conventional dentures, the insertion of removable implant-supported dentures did not improve the nutritional status of elderly patients yet increased the probability of consuming fresh fruit and vegetables [55]. These considerations suggest that RDPs supported on implants are the option of choice in implant dentistry in a geriatric context; these concepts further boast the advantages of easy reconstruction in case of implant loss and lower costs. The latter is particularly true in constructions where implant-supported dentures are designed as overdentures and retained by stud attachments. However, wear and deterioration of inserts are frequently reported issues in implant-supported overdentures [56,57,58]. With regard to this aspect, laboratory data indicate that inserts fabricated from polyaryletherketones feature superior mechanical properties [59,60]; clinical data are, however, missing to date. Ball attachments are commonly employed for retaining overdentures supported by single mandibular implants, yet clinical studies have shown that this treatment option coincides with relevant prosthetic aftercare due to wear and deterioration of the retentive elements [61]. Other retention systems for implant-supported removable dentures include double crowns or bars, which, however, have the disadvantage of higher costs. Particularly for bars, oral hygiene is complex and can only be properly performed by skilled and functionally uncompromized patients, which is why bar-retained RDPs are not the treatment of choice in gerodontology.

In addition to conventional implants, mini dental implants (MDIs) are an option for minimally invasively supplying older patients with implant-supported overdentures. MDIs feature a reduced diameter, usually between 1.8 and 2.5 mm, and can be inserted flaplessly and under existing RDPs. In recent years, several clinical trials and reviews dealing with the insertion of MDIs have been published, which underlines the increasing relevance of this therapeutic approach. Enkling and co-workers reported that the interforaminal insertion of four MDIs in the edentulous mandible produced a significant increase in maximum voluntary bite force and chewing efficiency over a period of five years [62]. With regard to this aspect, several systematic reviews underline that MDIs feature promising survival rates [63,64,65]. However, it has been shown that in the mandible, failure of implants was by far lower than in the maxilla (4.89% vs. 31.7%) [63]. With regard to this aspect, it has to be taken into consideration that MDIs are single-piece implants with fixed ball superstructure. In contrast to two-piece implants, the superstructure cannot be exchanged against cover screws, which might be a problem in geriatric patients that are reluctant or unable to wear the corresponding overdenture as the uncovered ball superstructure might produce irritations of soft tissues.

### 4.3. Duplicate Dentures

Due to cognitive and motoric impairment, adaptation toward novel complete dentures can be difficult for older patients. In these cases, the fabrication of duplicate dentures can be helpful, as relevant properties of the existing dentures such as the vertical or horizontal dimension of occlusion, phonetics, or esthetics can simply be transferred into novel dental prostheses. While the conventional approach employs impression techniques to produce a duplicate that can be used as impression tray and, subsequently, as a template for the arrangement of denture teeth, recent innovations in digital dentistry allow digitalization of existing dentures and processing of datasets for the fabrication of duplicate dentures with CAD/CAM-assisted subtractive or additive means [66]. With regard to this aspect, it is likely that digital techniques will help to spread the application of concept of duplicate dentures in a geriatric concept by reducing laboratory costs. Digitally fabricated complete dentures feature the advantages of improved quality of the processed materials with less accumulation of biofilms and easy replication in case of denture loss or fracture [67]. Judging from the limited evidence that is currently available, relevant concerns affecting up to 25% of reported cases relate to patient dissatisfaction, denture retention, and esthetic appearance [68,69]. Ellis and co-workers highlighted that the insertion of dentures fabricated in a conventional approach and with the duplication technique both increase oral health-related quality of life and patient satisfaction, although patients were less satisfied with the esthetic appearance of duplicate dentures [70]. For people with dementia, it might make sense to keep the denture and its fabrication process simple, i.e., low-cost duplicates fabricated from a simple material as a reserve in case the RDPs have been misplaced. Figure 1 displays a flowchart that illustrates the advantages and disadvantages associated with application of duplicate dentures.

## 5. Conclusions

Prosthetic treatment in geriatric patients should be carefully planned, taking into consideration that changes in general health may rapidly occur. Thus, the range of treatment options available is frequently limited, which underlines that the focus should be set on simple, stable, and solid prosthetic concepts. With regard to this aspect, it must be regarded that not only dentists and patients can be involved in the daily dental care in frail older patients, but also relatives, nurses, caring staff, and physicians. For planning prosthetic treatment in geriatric patients, OFC should be assessed as a basic instrument to identify the individual treatment options and any decision for prosthetic treatment should rest upon a participatory decision-making process.

## Figures and Tables

**Figure 1 jcm-10-00304-f001:**
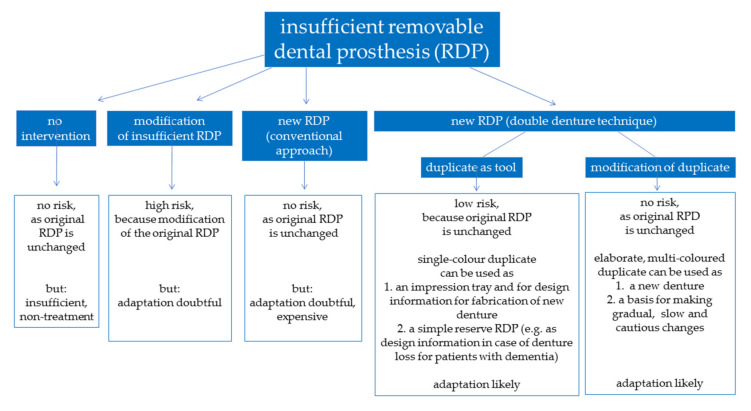
Prosthetic treatment approaches in edentulous patients and potential complications associated with adaptation to new RDPs.

**Table 1 jcm-10-00304-t001:** Description of the oral functional capacity (OFC) consisting of four resilience capacity levels (RCL 1—RCL 4) and three parameters (therapeutic capability, oral hygiene ability, self-responsibility). The parameter with the worst evaluation is used to classify the patients into one of the resilience capacity levels (RCL) [19,20].

Resilience Capacity Level (RCL)	Therapeutic Capability	Oral Hygiene Ability	Self-Responsibility
RCL 1 normal	normal	normal	normal
RCL 2 slightly reduced	slightly reduced	slightly reduced	
RCL 3 greatly reduced	greatly reduced	greatly reduced	reduced
RCL 4 no resilience	none	none	none

**Table 2 jcm-10-00304-t002:** Criteria that influence the assessment of the individual three parameters determining OFC (therapeutic capability, oral hygiene ability, self-responsibility).

Therapeutic Capability	Oral Hygiene Ability	Self-Responsibility
Treatment location	Gripping ability	Recognition of problems
Transportability	Hand grip strength	Expression of will
Transfer to the dental chair possible	Manual ability to clean teeth	Decision-making capability
Restrictions on patient positioning	Visual acuity	Uptake of dental services
Feasibility of diagnostic procedures	Performing oral hygiene procedures	Organizational skills/coordination
Ability to tolerate prolonged periods of mouth opening	Degree of difficulty in cleaning the oral cavity	After-care competence
Risk of medical incidents	Understanding of instructions/facts	(Legal) representative
Risk of drug interactions	Implementing advice received	
Risk during dental procedures	After-care competence	
Understanding of instructions/facts	Third person available to carry out oral hygiene	
After-care competence	Ability to purchase oral hygiene products unaided	
Manual dexterity		
Capability to adapt to new or modified denture		

## Data Availability

Data sharing is not applicable to this article.

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
