# Peer review of "Considerations for the Prosthetic Dental Treatment of Geriatric Patients in Germany"

_jcm, 2021, doi:10.3390/jcm10020304_

Round 1

Reviewer 1 Report

The present study entitled 'Considerations for the prosthetic dental treatment of geriatric patients' aims to provide a narrative summary outlining current pathways in geriatric dentistry from a prosthetic perspective. 

Background: The epidemiology of dental status in older people with particular focus on the German situation is similar to that in other industrialized societies. The overview is detailed but lack comments on the dental health care services offered for the older part of the German population. How is it organized?

Since edentulism decreases and people are getting older over time, need for dental care will be more frequently required. This might also imply more complex prosthetic solutions. It is recognized that conventional diagnostic means should be completed? (or complemented?) by specific tools derived from geriatric assessment. An integration of individuals' oral functional capacity is recommended. This is fine - but a further development of this approach together with participatory decision making, also recommended, could have been further elaborated in this article. 

Authors proceed by describing SDA , implant dentistry and duplicate dentures - however it is difficult to see the connection to the recommended complementary approach including OFC and participatory decision making.

In such the whole manuscript lacks focus. 

Some new concepts that are introduced are followed up by a more detailed description - such as the oral functional status (OFC) and shortened dental arch (STD). Other concepts are only introduced - without further explanation- such as participatory decision making. This gives an imbalanced impression  

Author Response

Dear reviewer, 

thank you for reviewing the manuscript "Considerations for the prosthetic dental treatment of geriatric patients". We carefully revised the manuscript in accordance with your recommendations and thank you for the opportunity to submit a revised version of the manuscript. Detailed comments are appended in the cover letter. 

Yours sincerely,

The authors of jcm-1055241

Reviewer 2 Report

The authors exhibited a exaustive narrative review . I think the information is interesting and would be useful to the readership. However, before this article can be accepted for publication, I suggest the authors revise the manuscript as follows:

  1. Title : Please add more detailed informations related to the specific group of old patients examined ( European /German patients) in the present review.
  2. Line n. 142 : provide more informations on different RPD’s solutions and if there are significant correlations between different RDP’s, their economic cost, and incidence of biological and/or mechanical complications ( eg. caries on abutments, loss of retention, metal/resin fracture of RDP)
  3. Line n. 170 : provide more informations on updated treatment plan concepts for a rationale prosthetic planning. The treatment plan is the key of all prosthetic solutions; underline the outcomes of the most important and related long termed clinical studies.
  4. Line n. 222 : Implant dentistry: it not sufficient to describe the various solutions of implant-supported /retained prostheses. Please, add adequate information on removable implant retained/supported solutions ( eg. Overdenture with ball attachments or with bar ) and fixed implant-supported solutions (eg. screw retained protheses as Toronto bridge). Integrate adequate references to evaluate these treatment options in terms of success rate and survival rate.
  5. Line n. 248 : please add references for the 3-S principle
  6. Line 293 : Duplicate dentures: add more information on the digital denture, the most recent frontier in digital dentistry. What are the clinical outcome of this solution? Please, underline advantages and disadvantages in terms of quality of this solution (denture retention, stability), based on the limited clinical studies present in literature.

Author Response

(The authors gave the same response as above.)
